# Investigation of the Frequency of Detection of Common Respiratory Pathogens in Nasal Secretions and Environment of Healthy Sport Horses Attending a Multi-Week Show Event during the Summer Months

**DOI:** 10.3390/v15061225

**Published:** 2023-05-24

**Authors:** Nicola Pusterla, Madalyn Kalscheur, Duncan Peters, Lori Bidwell, Sara Holtz, Samantha Barnum, Kaila Lawton, Matt Morrissey, Stephen Schumacher

**Affiliations:** 1Department of Medicine and Epidemiology, School of Veterinary Medicine, University of California, Davis, CA 95616, USA; smmapes@ucdavis.edu (S.B.); kolawton@ucdavis.edu (K.L.); 2East–West Equine Sports Medicine, Lexington, KY 40583, USA; kals0032@umn.edu (M.K.); dpetersdvm1@gmail.com (D.P.); loribidwell@hotmail.com (L.B.); sarajholtz@gmail.com (S.H.); 3Morrissey Management Group, LLC, Wellington, FL 33414, USA; matt@mmg.management; 4US Equestrian Federation, Lexington, KY 40511, USA; sschumacher@usef.org

**Keywords:** respiratory pathogens, surveillance, nasal swabs, environmental samples, qPCR, healthy sport horses

## Abstract

Little information is presently available regarding the frequency of the silent shedders of respiratory viruses in healthy sport horses and their impact on environmental contamination. Therefore, the aim of this study was to investigate the detection frequency of selected respiratory pathogens in nasal secretions and environmental stall samples of sport horses attending a multi-week equestrian event during the summer months. Six out of fifteen tents were randomly selected for the study with approximately 20 horse/stall pairs being sampled on a weekly basis. Following weekly collection for a total of 11 weeks, all samples were tested for the presence of common respiratory pathogens (EIV, EHV-1, EHV-4, ERAV, ERBV, and *Streptococcus equi* ss *equi* (*S. equi*)) using qPCR. A total of 19/682 nasal swabs (2.8%) and 28/1288 environmental stall sponges (2.2%) tested qPCR-positive for common respiratory pathogens. ERBV was the most common respiratory virus (17 nasal swabs, 28 stall sponges) detected, followed by EHV-4 (1 nasal swab) and *S. equi* (1 nasal swab). EIV, EHV-1, EHV-4 and ERAV were not detected in any of the study horses or stalls. Only one horse and one stall tested qPCR-positive for ERBV on two consecutive weeks. All the other qPCR-positive sample results were related to individual time points. Furthermore, only one horse/stall pair tested qPCR-positive for ERBV at a single time point. The study results showed that in a selected population of sport horses attending a multi-week equestrian event in the summer, the frequency of the shedding of respiratory viruses was low and primarily restricted to ERBV with little evidence of active transmission and environmental contamination.

## 1. Introduction

With the increased concerns about respiratory outbreaks, especially equine herpesvirus-1 (EHV-1), many equine showgrounds are struggling to institute compliant protocols with the goal to reduce the risk of transmission. Vaccination requirements, health certificate, biosecurity protocols, daily physical monitoring and pre-show testing are all measures that have been used in the past to reduce the risk of disease transmission at equestrian events. Unfortunately, there is very little contemporary information on the detection frequency of circulating respiratory pathogens in healthy adult horses. Collectively, various studies have shown that the detection frequency of selected respiratory pathogens ranges from 0 to 4% depending on the time of the year, and age and use of the population tested [1,2,3,4,5,6,7,8,9]. Further, no environmental program for respiratory pathogens has been validated to date, in order to indirectly monitor the presence and spread of such pathogens. This information is essential in order to institute scientifically sound protocols with an aim to monitor the populations of healthy show horses and reduce the risk of disease spread. Therefore, the aim of this study was to investigate the detection frequency of common respiratory pathogens in nasal secretions and stalls of sport horses attending a multi-week equestrian event during the summer months.

## 2. Materials and Methods

### 2.1. Study Location and Population

The study was performed at a large multi-week horse show located in Williamsburg, MI, USA (Traverse City Horse Shows; https://traversecityhorseshows.com, accessed on 6 June 2022) during the summer of 2022. 

To allow consistency in sample collection, the investigators selected approximately 120 horse/stall pairs across 6/15 randomly selected tents. Because of the voluntary nature of the study, the selection of horses and stalls within each tent could not be randomized and selection was based on the willingness of trainers/owners to participate in the study. Written owner’s consent was acquired prior to any sample collection. The temporary tents and stalls were set up for the show and the stalls were cleaned and disinfected prior to being used.

### 2.2. Sample Collection

Rostral nasal swabs and environmental stall samples were collected weekly for 11 weeks from every horse/stall pair enrolled in the study. Wearing disposable gloves, the attending veterinarian collected nasal secretions from the rostral nasal passages using a 6″ rayon-tipped swab (Puritan^®^, Guilford, ME, USA). The swabs were placed in 10 mL evacuated blood tubes (BD Vacutainer^®^, Franklin Lakes, NJ, USA) without any added solution and labeled with date, name of horse, barn and stall number. Stall samples were collected using sponges soaked in a neutralizing buffer (3M Sponge-Stick with 10 mL Neutralizing Buffer, St. Paul, MN, USA). Collection of environmental samples included the swabbing of the inside of the stall door, stall walls, and rim of the feeder and/or water bucket, if available. In order to prevent possible cross-contamination during sample collection and handling, the collection of stall sponges followed stringent biosecurity protocols. If a horse was not present at the time of stall sample collection, only the environmental sample was collected. Nasal swabs and stall sponges were kept refrigerated overnight and shipped next day on ice to the laboratory for sample processing and analysis. 

### 2.3. Sample Analysis

Nucleic acid extraction from nasal swabs and stall sponges was performed using an automated nucleic acid extraction system (QIAcube HT, Qiagen, Valencia, CA, USA) according to the manufacturer’s recommendations. The swabs were assayed for the presence of equine influenza virus (EIV), equine herpesvirus-1 (EHV-1), EHV-4, equine rhinitis A virus (ERAV), equine rhinitis B virus (ERBV), and *Streptococcus equi* ss. *equi* (*S. equi*) using previously reported real-time TaqMan PCR assays [10,11,12].

Frequency of respiratory pathogen detection from the study horses and stalls was evaluated using descriptive analyses. 

## 3. Results

A total of 682 nasal swabs and 1288 environmental stall sponges were collected throughout the study period. The number of weekly collected nasal swabs ranged from 0 to 23 swabs (median of 11.5 swabs) per tent, while the number of weekly collected stall sponges ranged from 0 to 25 (median of 22 swabs) per tent. A total of 19/682 (2.8%) nasal swabs tested qPCR-positive for one of the selected common respiratory pathogens with 17 ERBV, 1 EHV-4 and 1 *S. equi* qPCR-positive nasal swabs (Table 1). The majority of the horses tested qPCR-positive for a respiratory pathogen at one single time point, while one horse tested qPCR-positive for ERBV in two consecutive weeks. The percentage of qPCR-positive nasal swabs per week ranged from 0 to 8.1% (median 1.6%) with the highest frequency of qPCR-positive nasal swabs recorded during the last two study weeks (7.4% on week 10 and 8.1% on week 11). The detection frequency of respiratory pathogens by tent ranged from 1.1 to 4% with the highest detection rate observed for Tent 9 (Table 2). A total of 28/1288 (2.2%) environmental stall sponges tested qPCR-positive for ERBV (Table 1). The majority of the qPCR-positive stalls tested positive for ERBV at a single time point, while one stall tested qPCR-positive for ERBV in two consecutive weeks. The percentage of ERBV qPCR-positive stall sponges per week ranged from 0 to 4.2% (median 1.7%) with the highest frequency of qPCR-positive stall sponges recorded during week 8 of the study. The detection frequency of respiratory pathogens by tent ranged from 0 to 4.3% with the highest detection rate observed for Tent 9 (Table 2). When nasal swabs and environmental stall sponges were compared for the same horse/stall pairs, only one single pair tested qPCR-positive for ERBV at the same time point. The horse/stall pair was located in Tent 9.

## 4. Discussion

The majority of outbreaks associated with respiratory pathogens in performance horses attending large equestrian events have often been reported during the colder months of the year [13,14,15,16,17]. Outbreaks with respiratory pathogens seldom occur during the summer months, an observation that may relate to environmental factors (high ambient temperature and low humidity) reducing pathogen viability and to a low frequency of respiratory viruses and bacteria circulating amongst horses [18]. The present study results showed that the shedding of well-characterized equine respiratory pathogens was an uncommon finding, as only 19 out of 682 nasal swabs collected over an 11-week period tested qPCR-positive, with ERBV being the predominant virus detected. Further, environmental contamination was infrequent and restricted to ERBV with no evidence of spread.

While ERBV is endemic in horse populations based on seroprevalence studies, natural infection has been reported in horses displaying fever, anorexia, seromucoid nasal discharge, coughing, lymphadenopathy and occasionally lower limb swelling [12,18]. Detection rates of ERBV qPCR-positive nasal swabs from horses with fever and respiratory signs has been reported to range from 2.3 to 5.1% [12,18,19]. Interestingly, clinical ERBV infections in horses have shown no seasonality; a greater frequency in horses less than 1 year of age and in horses used for competition [18]. Subclinical ERBV infection has also been reported with frequencies ranging from 0.8 to 1.3% depending on the studied population [17,18,20]. The clinical relevance of ERBV detection in nasal secretions from the present sport horse population still has to be determined, as respiratory disease was not reported by show veterinarians during the show period. 

The study population was composed of adult sport horses competing over an 11-week period. One would have expected the detection frequency of respiratory pathogens to increase over time in both nasal secretions and environmental samples. The detection frequency of ERBV in nasal secretions (2.8%) and stalls (2.2%) was comparable, although twice as many environmental stall sponges were tested compared to nasal swabs. Peak detection frequency of ERBV in nasal secretions was documented during the last two study weeks, potentially reflecting increased subclinical transmission rate overtime. The highest frequency of ERBV detection in stalls was observed during week 5 and linked to a cluster of three stalls in Tent 9. It is interesting to note that Tent 9 displayed the highest detection rate of ERBV in nasal secretions and stalls when compared to the other tents. This observation may relate to the increased shedding and environmental contamination within the horses housed in Tent 9. Two recent studies have been reported on the detection of ERBV from the feces of healthy sport horses and horses with diarrhea or weight loss [9,21]. While the presence of common respiratory pathogens, such as EIV, EHV-1, EHV-4 and *S. equi*, is not to be expected in the fecal material, the acid-stable ERBV can be found in feces. It is therefore not possible to determine if the origin of the ERBV qPCR-positive stall sponges came from nasal secretions and/or feces, as fecal samples were not tested for common respiratory pathogens. With the exception of one single ERBV qPCR-positive horse/stall pair, all other ERBV positive samples were not temporally or geographically related. This observation likely relates to the short shedding period of ERBV infected horses and the short detection time of RNA viruses in the environment [20,22]. This is further supported by the lack of longitudinal detection of ERBV in the same horse or the same stall. While qPCR testing of individual nasal swabs for respiratory viruses and bacteria provides an insight into active shedding, environmental stall samples reflect past and present shedding and represent a more accurate measure of pathogen buildup over time. Other respiratory pathogens, especially EHV-4 and *S. equi*, were sporadically detected in the study population. The very low detection rate of these respiratory pathogens in the sport horses might have been related to various host and environmental factors, which reduced infection, transmission, shedding, environmental contamination and pathogen survival. Summer has been linked to a low incidence of respiratory infections in both clinically and subclinically infected equids [18,20]. From a practical standpoint, the regular testing of individual or pooled environmental stall samples for respiratory pathogens by qPCR could be an additional biosecurity step intended to monitor environment spread over time. 

Study limitations related to the nature of the convenience study, which was based on the voluntary enrollment of the sport horses. Random enrollment was not possible, as written owner’s consent was required. Another limitation was that the nasal swabs were only matched to half of the corresponding stall sponges, as many horses were not in their stalls at the time of the weekly sample collection. Furthermore, based on previous reports that ERBV can be detected in the feces, the concurrent collection and analysis of feces would have been relevant for this study. However, independent of the environmental origin of ERBV, this virus was successfully used as a marker for transmission and environmental contamination in this population of sport horses. 

## 5. Conclusions

In conclusion, the study results showed that in a selected population of sport horses attending a multi-week equestrian event during the summer, the shedding frequency of respiratory viruses was low and primarily restricted to ERBV with little evidence of active transmission and environmental contamination. The data supports previous observations that respiratory pathogens circulate at a low rate amongst sport horses during the summer months, making an occurrence of outbreaks less likely. However, one needs to keep in mind that the present results are specific to the studied horse population and reflect environmental factors specific to the study period. It is also important to highlight that while nasal swabs and environmental samples allow to assess the presence of respiratory pathogens circulating amongst at risk horses, such a strategy is not intended to replace good biosecurity protocols.

## Figures and Tables

**Table 1 viruses-15-01225-t001:** Weekly detection of respiratory pathogens by qPCR in nasal swabs and stall sponges collected from healthy sport horses during a multi-week equestrian event.

Week	Nasal Swab		Stall Sponge	
	qPCR Positive/Total Samples	Pathogen	qPCR Positive/Total Samples	Pathogen
Week 1	1/96	ERBV (1)	1/117	ERBV (1)
Week 2	1/80	ERBV (1)	3/116	ERBV (3)
Week 3	1/36	ERBV (1)	2/116	ERBV (2)
Week 4	1/67	ERBV (1)	4/123	ERBV (4)
Week 5	2/83	ERBV (2)	3/129	ERBV (3)
Week 6	1/62	S. equi (1)	5/132	ERBV (5)
Week 7	3/79	ERBV (2), EHV-4 (1)	0/119	No pathogen detected
Week 8	0/48	No pathogen detected	3/72	ERBV (3)
Week 9	0/15	No pathogen detected	3/120	ERBV (3)
Week 10	4/54	ERBV (4)	2/124	ERBV (2)
Week 11	5/62	ERBV (5)	2/120	ERBV (2)
Total	19/682 (2.8%)		28/1288 (2.2%)	

**Table 2 viruses-15-01225-t002:** Frequency of detection of respiratory pathogens in nasal secretions and environmental stall sponges collected from healthy sport horses during a multi-week equestrian event. The results are listed by sample type and housing location.

	Tent 1	Tent 3	Tent 6	Tent 9	Tent 10	Tent Imperial
Nasal swab	6/154 (3.9%)	1/81 (1.2%)	1/92 (1.1%)	8/200 (4.0%)	2/83 (2.4%)	1/72 (1.4%)
Stall sponge	9/258 (3.5%)	3/260 (1.1 %)	2/156 (1.3%)	11/256 (4.3%)	3/218 (1.4%)	0/140 (0%)

## Data Availability

Not applicable.

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
