# Peer review of "Investigation of the Frequency of Detection of Common Respiratory Pathogens in Nasal Secretions and Environment of Healthy Sport Horses Attending a Multi-Week Show Event during the Summer Months"

_viruses, 2023, doi:10.3390/v15061225_

Round 1

Reviewer 1 Report

Comments

In the manuscript, but particularly in the Abstract, consider changing the order that the pathogens are presented to that of clinical importance/concern, e.g. EIV and EHV first, to emphasize your findings.

Lines 75/76 the "kit" from 3M has a brand name, please use that, i.e. Swab-sampler with neutralizing buffer, to make clear that you are using an available "kit"

Line 82 when were the samples shipped in relation to being taken / how long were they kept refrigerated prior to shipping

The fact that in general the nasal and environmental swabs yielded quite similar results is interesting and may merit further discussion. If there is a sufficient body of evidence on this could one be done rather than the other and if so which.

Lines 144 and 152-156 Is the age of the horses known? Given that an adult horse could have an age anywhere between 4 and 20 the actual age of the horse might be significant. Is there a way of matching age of horse to positive/negative samples.

Line 171 Is "negatively impacted" the correct terminology: surely this is a positive impact if there is reduced infection, transmission, shedding, environmental contamination and pathogen survival. Even if you do not know which of these factors actually played a role, consider using another word e.g. "reduce"

Lines 175-177 What does this imply? For example, are owners of older or "healthier" horses more likely to agree to participate and thus you are finding fewer positive results? Please consider and discuss the impact of this clearly described bias/limitation

Editorial comments

Line 21 insert "the" before " summer months"

Line 32 amend to "frequency of shedding"

Line 39 write equine herpesvirus in full the first time it is mentioned

Line  102 propose rewording to "horses that tested positive (qPCR) for a pathogen"

Line 103, 106, 110 "in consecutive weeks" rather than "on"

Line 143 "during" rather than "through"

Line 160 "with" rather than "at"

Line 183 "this" rather than "the"

The authors are to be congratulated for conducting a well designed study and for presenting a well written manuscript. The manuscript was a pleasure to read.

Author Response

Reviewer 1:

In the manuscript, but particularly in the Abstract, consider changing the order that the pathogens are presented to that of clinical importance/concern, e.g. EIV and EHV first, to emphasize your findings.

The order of the pathogens has been changed as suggested by the reviewer.

Lines 75/76 the "kit" from 3M has a brand name, please use that, i.e. Swab-sampler with neutralizing buffer, to make clear that you are using an available "kit"

 The name of the kit has been included in line 75/76.

Line 82 when were the samples shipped in relation to being taken / how long were they kept refrigerated prior to shipping

The samples were kept refrigerated overnight shipped the next day so to arrive to the laboratory within 24 hours of shipment. Additional information has been added in the manuscript.

The fact that in general the nasal and environmental swabs yielded quite similar results is interesting and may merit further discussion. If there is a sufficient body of evidence on this could one be done rather than the other and if so which.

The reviewer brings up a very important point, which warrants further discussion. The regular nasal swabbing of horses at a show in order to detect silent shedders is not practical, logistically impossible and financially unsustainable. However, knowing that environmental stall samples reflect past and present nasal shedding seems to offer a fair alternative. One possible application in the future could be the regular swabbing of stalls, which could be done in pools (i.e. one sponge for every 5-10 stalls)  in order to monitor environmental contamination with contagious respiratory pathogens. It is mostly the clustering of positive stalls that would trigger further restrictions or testing. Additional information has been added in order to expand on this topic.

Lines 144 and 152-156 Is the age of the horses known? Given that an adult horse could have an age anywhere between 4 and 20 the actual age of the horse might be significant. Is there a way of matching age of horse to positive/negative samples.

Unfortunately, the specific age of the study horses was not recorded for this study. According to the show organizer, the age of the show horses attending the event ranged from 4 to 20 years.

Line 171 Is "negatively impacted" the correct terminology: surely this is a positive impact if there is reduced infection, transmission, shedding, environmental contamination and pathogen survival. Even if you do not know which of these factors actually played a role, consider using another word e.g. "reduce"

The authors fully agree with the reviewer about the terminology and the term “reduced” was used instead of “negatively impacted”.

Lines 175-177 What does this imply? For example, are owners of older or "healthier" horses more likely to agree to participate and thus you are finding fewer positive results? Please consider and discuss the impact of this clearly described bias/limitation

Most clinical studies using healthy horses are often based on the willingness of horse owner to participate. While the study only included healthy sport horses, the authors don’t believe that age played a role in enrollment. The timing of the study was pertinent as the year prior to the study, many large outbreaks of EHM occurred at equestrian venues. Hence, the owners/trainers/agents were all aware of the situation and most willing to participate in the study. The most common deterrent for not participating in the study was the known refractory nature of some horses to nasal swab collection.

Editorial comments

Line 21 insert "the" before " summer months"

Change made as requested.

Line 32 amend to "frequency of shedding"

Change made as requested.

Line 39 write equine herpesvirus in full the first time it is mentioned

Change made as requested.

Line  102 propose rewording to "horses that tested positive (qPCR) for a pathogen"

Change made as requested.

Line 103, 106, 110 "in consecutive weeks" rather than "on"

Change made as requested.

Line 143 "during" rather than "through"

Change made as requested.

Line 160 "with" rather than "at"

Change made as requested.

Line 183 "this" rather than "the"

Change made as requested.

Reviewer 2 Report

Manuscript is well written and study design, methods and conclusions clearly stated.  Limits of study also presented well.  Some minor editing:

1. Line 28 -- EHV-1/-4 should be just EHV-1 as 4 was found.

2.  Line 101-102 and 108-109 are redundant.

3. Could provide age range of sampled horses if known.

Author Response

Reviewer 2:

Manuscript is well written and study design, methods and conclusions clearly stated.  Limits of study also presented well.  Some minor editing:

  1. Line 28 -- EHV-1/-4 should be just EHV-1 as 4 was found.

The suggested change was made in the manuscript.

  1. Line 101-102 and 108-109 are redundant.

While both sentences appear redundant, the first one applies to horses while the second one relates to stall sponges.

  1. Could provide age range of sampled horses if known

Unfortunately, the specific age of the study horses was not recorded for this study. According to the show organizer, the age of the show horses attending the event ranged from 4 to 20 years.

Reviewer 3 Report

It would be desirable if the 95% confidence intervals for the frequency percentages were also included  (for proportions it would be sqrt [p(1-p)/n] )

Author Response

Reviewer 3:

It would be desirable if the 95% confidence intervals for the frequency percentages were also included  (for proportions it would be sqrt [p(1-p)/n] ).

A confidence interval is a range of estimates for an unknown parameter. In the present study the frequency of detection for the various respiratory pathogens was calculated based on the number of qPCR-positive results for a specific pathogen divided by the entire number of either swabs or stall sponges collected. In this instance the population (horses and sponges) was well known.